# Activation of the Gut–Brain Interaction by Urolithin A and Its Molecular Basis

**DOI:** 10.3390/nu16193369

**Published:** 2024-10-03

**Authors:** Daiki Kubota, Momoka Sato, Miyako Udono, Akiko Kohara, Masatake Kudoh, Yuichi Ukawa, Kiichiro Teruya, Yoshinori Katakura

**Affiliations:** 1Graduate School of Bioresources, Bioenvironmental Sciences, Kyushu University, Fukuoka 819-0395, Japan; kubota.daiki.441@s.kyushu-u.ac.jp (D.K.); sato.momoka.023@s.kyushu-u.ac.jp (M.S.); 2Faculty of Agriculture, Kyushu University, Fukuoka 819-0395, Japan; mudono@grt.kyushu-u.ac.jp (M.U.); kteruya@grt.kyushu-u.ac.jp (K.T.); 3Daicel Corporation, Tokyo 108-8230, Japanms_kudoh@jp.daicel.com (M.K.); yi_ukawa@jp.daicel.com (Y.U.)

**Keywords:** urolithin A, gut–brain interaction, exosome

## Abstract

**Background:** Urolithin A (Uro-A), a type of polyphenol derived from pomegranate, is known to improve memory function when ingested, in addition to its direct effect on the skin epidermal cells through the activation of longevity gene SIRT1. However, the molI ecular mechanism by which orally ingested Uro-A inhibits cognitive decline via the intestine remains unexplored. **Objectives:** This study aimed to evaluate the role of Uro-A in improving cognitive function via improved intestinal function and the effect of Uro-A on the inflammation levels and gene expression in hippocampus. **Methods:** Research to clarify the molecular basis of the functionality of Uro-A was also conducted. **Results:** The results demonstrated that Uro-A suppressed age-related memory impairment in Aged mice (C57BL/6J Jcl, male, 83 weeks old) by reducing inflammation and altering hippocampal gene expression. Furthermore, exosomes derived from intestinal cells treated with Uro-A and from the serum of Aged mice fed with Uro-A both activated neuronal cells, suggesting that exosomes are promising candidates as mediators of the Uro-A-induced activation of gut–brain interactions. Additionally, neurotrophic factors secreted from intestinal cells may contribute to the Uro-A-induced activation of gut–brain interactions. **Conclusions:** This study suggests that Uro-A suppresses age-related cognitive decline and that exosomes and other secreted factors may contribute to the activation of the gut–brain interaction. These findings provide new insights into the therapeutic potential of Uro-A for cognitive health.

## 1. Introduction

Urolithin A (Uro-A) is a gut microbiota-derived metabolite of ellagitannins and ellagic acid found in pomegranates, strawberries, and raspberries. Numerous functionalities of Uro-A have been reported, such as its effects on mitochondrial function and anti-inflammatory properties [1]. One of the most characteristic functions of Uro-A is the improvement in mitochondrial function. This is achieved through the activation of a selective autophagy process called mitophagy, in which Uro-A promotes the clearance and recycling of dysfunctional mitochondria. The process is believed to be a promising strategy for counteracting the functional decline in organs that occurs with aging. Recent clinical and animal studies have demonstrated that Uro-A exerts anti-inflammatory effects. In particular, it has been reported that Uro-A treatment suppresses inflammation in the brain of Alzheimer’s disease model mice and can improve their cognitive function [2,3]. Furthermore, it has been reported that Uro-A can extend the lifespan of model organisms, such as Drosophila and mice, inhibit age-related muscle dysfunction, suppress inflammatory bowel disease, and have protective effects against metabolic disorders, such as dyslipidemia, obesity, and glucose intolerance [1]. Although the direct effects of Uro-A on the skin epidermal cells are known [4], the mechanism by which orally administered Uro-A affects external organs indirectly via the intestine remains unexplored.

This study aimed to elucidate the function of Uro-A, a multifunctional polyphenol, as a food source. Specifically, the indirect function of Uro-A in improving brain function via the intestine was investigated by examining the effect of Uro-A on the suppression of cognitive decline observed in Aged mice. This study also aimed to clarify the molecular mechanisms of Uro-A functionality.

## 2. Materials and Methods

### 2.1. Cell Culture and Reagent

The human colon cancer cell line Caco-2 (ATCC, Manassas, VA, USA) and the human neuronal cell line SH-SY5Y (ATCC) were cultured in Dulbecco’s Modified Eagle Medium (DMEM) (Nissui, Tokyo, Japan) containing 10% heat-inactivated fetal bovine serum (FBS) (Capricorn Scientific GmbH, Ebsdorfergrund, Germany) at 37 °C in 5% CO_2_. Uro-A was purchased from Tokyo Chemical Industory Co., Ltd. (Tokyo, Japan).

### 2.2. Mitochondria

Cells were stained with 250 nM MitoTracker Red CMXRos (Thermo Fischer Scientific, Waltham, MA, USA) at 37 °C for 30 min and then with Hoechst 33,342 (Dojindo, Kumamoto, Japan) at 37 °C for 30 min. Stained cells were analyzed using an IN Cell Analyzer 2200 (Cytiva, Tokyo, Japan) to quantitatively determine the number of cells with activated mitochondria using IN Cell Investigator high-content image analysis software version 1.3 (Cytiva).

### 2.3. Quantitative Reverse Transcriptase–Polymerase Chain Reaction (RT-qPCR)

RNA was prepared from cells using a High Pure RNA Isolation kit (Roche Diagnostics GmbH, Mannheim, Germany), as described previously [5]. RT-qPCR was performed using the GoTaq 1-Step RT-PCR System (Promega, Madison, WI, USA) and Thermal Cycler Dice Real-Time System TP-800 (Takara, Shiga, Japan). The samples were analyzed in triplicate. The PCR primer sequences used were as follows: human β-actin (ACTB) forward primer 5′-TGGCACCCAGCACAATGAA-3′ and reverse primer 5′-CTAAGTCATAGTCCGCCTAGAAGC-3′: human SIRT1 forward primer 5′-GCCTCACATGCAAGCTCTAGTG -3′ and reverse primer 5′-TTCGAGGATCTGTGCCAATCAT-3′: human SIRT3 forward primer 5′-AGCCCTCTTCATGTTCCGAAGTGT-3′ and reverse primer 5′-TCATGTCAACACCTGCAGTCCCTT-3′: human nicotinamide phosphoribosyltransferase (NAMPT) forward primer 5′-GGGTTACAAGTTGCTGCCACC-3′ and reverse primer 5′-GCAAACCTCCACCAGAACCG-3′: human brain-derived neurotrophic factor (BDNF) forward primer 5′-GTCAAGTTGGGAGCCTGAAATAGTG-3′ and reverse primer 5′-AGGATGCTGGTCCAAGTGGTG-3′: human PGC-1α forward primer 5′-GCTGACAGATGGAGACGTGA-3′ and reverse primer 5′-TAGCTGAGTGTTGGCTGGTG-3′: human neurotrophin-4 (NT-4) forward primer 5′-CCCGCTGCAAGGCTGATAAC-3′ and reverse primer 5′-CGCACATAGGACTGCTTGGC-3′: human ciliary neurotrophic factor (CNTF) forward primer 5′-ACCTTCCATGTTTTGTTGGC-3′ and reverse primer 5′-ATCTGGTATGCAAAGGCAGC-3′: human nerve growth factor (NGF) forward primer 5′-CAACAGGACTCACAGGAGCA-3′ and reverse primer 5′-AGGATGCTGGTCCAAGTGGTG-3′. β-actin was used as a housekeeping gene. Samples were normalized and analyzed using the ΔΔCt method [6].

### 2.4. Animals

All experiments were conducted according to the Guide for the Care and Use of Laboratory Animals and approved by the Ethics Committee on Animal Experimentation (Kyushu University; approval number: A22-271-3). The criteria for excluding animals from an experiment or analysis are as described in the animal experiment application form. Male C57BL/6J Jcl mice (30 weeks old) were provided by CLEA Japan (Tokyo, Japan). Aged mice (C57BL/6J Jcl, male, 83 weeks old) were provided by the Foundation of Biomedical Research and Innovation at Kobe through the National BioResource Project of MEXT, Japan. Referring to previous studies using Aged mice for a behavior test, male mice were used in the present study [7,8]. Sample size was determined on the basis of an initial investigation conducted to test memory functions. Ten mice per group were used and housed in cages of five mice per cage, maintained on a 12 h light/dark cycle, and allowed ad libitum access to food and water. The assignment of mice to experimental and control groups was random and conducted by a responsible person who was not in charge of the actual experiment. All behavioral procedures were conducted during the light phase of the cycle. The Aged Uro-A-fed (Aged-Uro-A) group was fed ad libitum for 56 d with a powdered MF diet (KBT Oriental Corporation, Saga, Japan) containing 0.16% pomegranate extract fermentation product powder and 11.3% Uro-A (Urorich) (Daicel Corporation, Osaka, Japan). The Young and the Aged-control (Aged-Ctrl) groups were fed a powdered MF diet mixed with maltodextrin ad libitum (Daicel Corporation, Osaka, Japan) for 56 d. The powdered diet was fed with Roden CAFÉ (Oriental Yeast Co., Ltd., Tokyo, Japan). To minimize potential confounders, treatments and measurements were randomized and carried out in double-blind studies. Weight loss exceeding 10% compared to control mice kept without food administration was considered a humane endpoint.

### 2.5. Novel Object Recognition Test (NORT)

To clarify the role of Uro-A in object recognition (OR) memory formation, NORT was performed in accordance with the method described previously [9]. For the habituation phase, individual adult male mice were placed in a chamber (40 cm × 40 cm × 30 cm) and allowed to explore freely for 10 min over 3 d. In the training phase, the mice were placed in the same chamber containing two different objects for 10 min and allowed to explore the objects. After 24 h, in the testing phase, one of the objects was exchanged for a new one, ensuring that the locations of the two objects were not changed. The duration of exploratory behavior exhibited by the mice was determined, and exploration preference was calculated. The discrimination index was calculated using the following formula:(time exploring the novel object − time exploring the familiar object)/(time exploring the novel object + time exploring the familiar object)

### 2.6. Fluorescent Immunocytochemistry

Brain samples were collected from the heads of the mice after NORT was completed and fixed in 10% formalin buffer. Paraffin-embedded brain sections were prepared as described previously [10]. For immunohistochemical analysis, the tissue sections were deparaffinized, rehydrated, and soaked in 1× HistoVT One (Nacalai Tesque, Kyoto, Japan). The sections were then heated at 90 °C for 20 min to activate the antigens. After washing with 0.1% Tween 20/TBS, the tissues were blocked with Blocking One Histo (Nacalai Tesque) for 1 h at room temperature. Brain sections were incubated with primary antibodies for anti-Iba1 (marker for microglia) (Cell Signaling Technology, Danvers, MA, USA), GFAP (marker for astrocytes) (Cell Signaling Technology), and NeuN (marker for neurons) (Biolegend, San Diego, CA, USA) and subsequently with secondary antibodies. Iba1 and GFAP were stained with Alexa Fluor 555 anti-rabbit IgG (Thermo Fischer Scientific) and NeuN with Alexa Fluor 488 anti-mouse IgG (Jackson ImmunoResearch, West Grove, PA, USA) [11].

After staining with Vectashield mounting medium (Vector Laboratories, Burlingame, CA, USA), the tissue samples were observed under a fluorescence microscope (EVOS M5000 Cell Imaging System, Thermo Fischer Scientific).

### 2.7. RNA Sequencing (RNAseq)

RNA was extracted from the hippocampus and cells using TRIzol Reagent (Thermo Fisher Scientific) and purified using the SV Total RNA Isolation System (Promega). RNA samples were quantified using an ND-1000 spectrophotometer (NanoDrop Technologies, Wilmington, DE, USA), and their quality was confirmed using a TapeStation (Agilent Technologies, Inc., Santa Clara, CA, USA). Sequencing libraries were prepared using the MGIEasy rRNA Depletion Kit and MGIE RNA Directional Library Prep Set (MGI Tech Co., Ltd., Shenzhen, China). The libraries were sequenced on a DNBSEQ-G400 FAST Sequencer (MGI Tech) by Bioengineering Lab (Kanagawa, Japan) and Cell Innovator (Fukuoka, Japan). Differential expression and pathway analyses were performed using the integrated iDEP web application (ver. 2.01, http://bioinformatics.sdstate.edu/idep/, accessed on 1 Februay 2024) [12]. The criteria for the differentially expressed genes were then established: *p*-value ≤ 0.05 and ratio ≥ 1.5-fold (upregulated genes) or 0.66-fold (downregulated genes).

### 2.8. Exosome Isolation

Caco-2 cells were cultured in DMEM containing 10% exosome-depleted FBS medium heat-inactivated supplement (System Bioscience, Moutain View, CA, USA) and 100 µM of Uro-A, and the cell supernatant was prepared. Mouse serum was prepared from the blood collected from the heart after euthanasia. The MagCapture Exosome Isolation Kit PS Ver. 2 (Fujifilm Wako Pure Chemical Corp, Osaka, Japan) was used to isolate exosomes from the cell supernatant and mouse serum [5,13]. The size distribution of the isolated exosomes was determined by dynamic light scattering using ELSZ-0S (Otsuka Electronics, Osaka, Japan) to ensure that no other particles were included. Concentrations of isolated exosome were expressed as protein equivalents determined by the micro BCA Protein Assay Kit (Thermo Fisher Scientific), and SH-SY5Y cells were treated with exosomes equivalent to 900 ng/mL proteins for 1 day.

### 2.9. miRNA Microarray Assay

The expression profiles of miRNAs in exosomes were evaluated using microarray analysis with a 3D-Gene Human and Mouse miRNA Oligo chip (Toray, Kanagawa, Japan). miRNA preparation and subsequent operations were outsourced to Kamakura Techno-Sciences, Inc. (Kanagawa, Japan). After global normalization of miRNA expression levels, the ratio of each gene was calculated for comparison between the control and experimental samples. The criteria for regulated genes were then established: (upregulated genes) ratio ≥ 2.0-fold [14]. The miRNA target genes were predicted using TargetScan (https://www.targetscan.org/vert_80/, accessed 3 March 2024). Tools and data provided by the Database for Annotation, Visualization, and Integrated Discovery (DAVID, https://david.ncifcrf.gov, accessed on 20 February 2024) were then used to determine significantly enriched pathways [15,16,17].

### 2.10. Statistical Analysis

For experiments that were repeated three times and when representative data were shown, they are described in the figure legend. The results are shown as the mean ± standard error. Multiple comparisons between groups were performed using a one-way ANOVA with Tukey’s post hoc test using KaleidaGraph 5 (Hulinks, Tokyo, Japan). Statistical significance was defined as *p* < 0.05 when compared to the control (* *p* < 0.05; ** *p* < 0.01; *** *p* < 0.001).

## 3. Results

### 3.1. Effects of Uro-A on Age-Related Memory Impairment in Aged Mice

Uro-A is known to improve memory function when ingested. However, the molecular mechanism by which orally ingested Uro-A inhibits cognitive decline via the intestine remains unexplored. Here, we tried to clarify the role of Uro-A in object recognition (OR) memory formation; we performed the NORT using Aged mice (Figure 1). The detailed protocol is described in the Materials and Methods Section. The results showed that the Young and Aged-Uro-A groups, but not the Aged-Ctrl group, explored the novel object significantly more than the familiar object during the test (Figure 2A). The results also revealed significant effects of Uro-A ingestion on the formation of OR memory in Aged mice, a trend similar to that observed in the Young group (Figure 2B). These results suggest that Uro-A ingestion ameliorated the decline in memory function in Aged mice.

### 3.2. Effects of Uro-A on Age-Related Inflammation in Aged Mice

Previous studies have indicated that glial cells such as microglia and astrocytes are the source of various mediators that significantly contribute to neuroinflammation, oxidative brain damage, and neuronal apoptosis [18,19,20]. Therefore, we used Iba-1 and GFAP as markers to verify the activation status of microglia and astrocytes. The results showed an increase in activated microglia and astrocytes in the dentate gyrus of the hippocampus of Aged-Ctrl mice compared to that in Young mice. Conversely, there was a significant decrease in activated microglia (Figure 3A,B) and astrocytes in Uro-A-fed mice (Figure 3A,C). Uro-A ingestion has been demonstrated to reduce the number of Iba-1-positive microglia and GFAP-positive astrocytes in the hippocampus of Aged mice. As previously reported [21], Iba1-positive microglia are required for object recognition memory. It is also known that glial cells can be distinguished into two states, resting and reactive states, and that glial cells in the reactive state are involved in pathogenesis and exhibit enlarged cell bodies [22]. Indeed, in Figure 3A, the hippocampus from Aged-Ctrl contains many Iba1-positive and enlarged cells, while such cells were reduced in the hippocampus from Aged-Uro-A, suggesting that the number of Iba1-positive microglia in the reactive state is reduced by Uro-A ingestion. Activated microglia are known to produce TNF-α and IL-1β neuroinflammatory cytokines. The enhanced production of these cytokines in Aged-Ctrl hippocampus and their attenuated production in Aged-Uro-A hippocampus supports the above results (Figure 4).

### 3.3. Effects of Uro-A on Gene Expression in the Hippocampus of Aged Mice

Next, RNA was prepared from mouse hippocampus and subjected to comprehensive gene expression analysis by RNAseq. The changes in gene expression in each mouse are shown as a heatmap in Figure 4A. The heatmap shows the results of the analysis of RNA from three mice mixed together for each subgroup. The genes were classified into four clusters (A–D) according to changes in gene expression. Pathways involving groups of genes belonging to each cluster are shown in Appendix A.

The results showed that the changes in gene expression in the hippocampi of Young and Aged-Ctrl mice were significantly different. In contrast, characteristic gene expression changes were also observed in Aged-Uro-A mice. In Aged-Uro-A mice, gene groups belonging to clusters A and D showed expression changes relatively close to those of Young mice. Furthermore, gene groups belonging to cluster B were identified as genes that showed Uro-A-specific expression changes, suggesting that Uro-A ingestion-specific changes were also found to be induced in the hippocampus.

KEGG pathway analysis revealed that Aged-Uro-A ingestion significantly regulated the expression of genes involved in the Neurotrophin signaling pathway in the hippocampus of Aged-Uro-A mice. In particular, the expression of BDNF and NT-3 was significantly increased (Appendix A). In addition, various factors that function intracellularly in response to signals from neurotrophic factors were significantly upregulated in the hippocampus of Uro-A-fed mice.

RNAseq analysis was used to verify the expression of mitochondria-related genes in the hippocampus of Uro-A-fed mice, and the expression of genes (SIRT1 [23], mitochondrial transcription factor A (TFAM) [24], Atp5d [25]) involved in mitochondrial biosynthesis and activity was significantly upregulated in the hippocampus of Uro-A-fed mice compared to Aged-Ctrl mice, suggesting that mitochondria are activated. Furthermore, RNAseq analysis of the expression of neurotrophic factors and cytokines involved in neuroinflammation revealed that BDNF expression, which decreased in the hippocampus of Aged mice, increased significantly with Uro-A ingestion, and that TNF-α and IL-1β expression, which increased in Aged mice [18,19,20], significantly decreased with Uro-A ingestion. These results suggest that Uro-A intake caused an increase in BDNF production in the hippocampus and a suppression of neuroinflammation.

### 3.4. Supernatants from Caco-2 Cells Treated with Uro-A-Activated SH-SY5Y Cells

In this study, we showed that Uro-A ingestion contributes to memory formation in Aged mice. Next, studies were conducted to elucidate the molecular basis for the activation of the gut–brain interaction by Uro-A. Then, we tested the possibility that Uro-A activates Caco-2 cells, which in turn activates SH-SY5Y cells via a factor secreted from activated Caco-2 cells. Caco-2 cells (1 × 10^4^ cells/well, 24-well plate, Corning, NY, USA) were treated with Uro-A for 2 days, and the culture supernatant was prepared. In total, 30% of the culture medium of SH-SY5Y cells (3.0 × 10^4^ cells/well, 96-well plate, Greiner bio-one, Tokyo, Japan) was replaced with the culture supernatant of Caco-2 cells and cultured for 1 day to verify whether SH-SY5Y is activated as an indicator of mitochondrial activation. The amount of culture supernatant added was determined based on the mitochondrial activation effect in SH-SY5Y cells associated with the addition of supernatant. No significant changes in the proliferation and viability of Caco-2 cells and SH-SY5Y cells during this treatment were observed. Uro-A was added in the range of 10–100 µM, which is commonly used as the concentration of polyphenols added. The residual Uro-A in the Caco-2 culture supernatant after 2 days of treatment with Uro-A was found to be below the detection limit [26]. The results of this study show that the culture supernatants derived from Caco-2 cells treated with Uro-A activated mitochondria in SH-SY5Y cells (Figure 5A) but not neurite outgrowth.

The activation was most pronounced when Uro-A was added at 100 µM, and this concentration was thus adopted as the concentration of Uro-A to be added in the following studies. Next, the effects of the supernatant from the Uro-A-treated Caco-2 cells on the expression of longevity genes, secreted factors, and factors involved in mitochondrial biogenesis in SH-SY5Y cells were evaluated. The longevity gene SIRT1 has been the focus of much attention in relation to mitochondrial biosynthesis [27]. SIRT3 is localized in mitochondria and is known to function in the regulation of reactive oxygen species levels [28]. Nicotinamide phosphoribosyltransferase (NAMPT) is known to be the rate-limiting enzyme that produces NMN from nicotinamide in the NAD^+^ salvage cycle, and its activation is believed to activate sirtuins through increased NAD^+^ [29]. BDNF plays an important role in maintaining synaptic plasticity in learning and memory [30]. PGC-1α is activated by SIRT1 through deacetylation and is believed to be involved in mitochondrial biogenesis [31]. The results showed that the Uro-A-treated Caco-2 supernatant significantly enhanced the expression of SIRT1, SIRT3, and NAMPT as longevity genes, BDNF, a secreted factor related to the brain–gut interaction, and PGC-1α, a mitochondria-related transcriptional coactivator in SH-SY5Y cells (Figure 5B–F), thus indicating that Uro-A treatment induces Caco-2 cells to secrete factors that can activate neuronal cells, thus consequently improving brain function.

### 3.5. Functional Evaluation of Uro-A-Induced Exosomes

Next, we focused on exosomes, known as extracellular vesicles, as the molecular basis for the activation of the gut–brain interaction by Uro-A, and examined their possible involvement. Therefore, we first isolated exosomes from the culture supernatants of Uro-A-treated Caco-2 cells and serum from Uro-A-fed Aged mice using the above kit. After each exosome was added to the SH-SY5Y cells and cultured, the neuronal cell activation potential of the exosomes was evaluated using intracellular mitochondrial activity as an indicator. The results showed that exosomes derived from Uro-A-treated Caco-2 cells activated mitochondria more strongly than those derived from untreated Caco-2 cells (Figure 6A). Similarly, exosomes derived from Uro-A-fed Aged mice activated mitochondria more strongly than those derived from Aged-Ctrl mice, and this activation was comparable to that derived from Young mice (Figure 6B). These results indicate a contribution of exosomes to the activation of the gut–brain interaction by Uro-A.

Next, by performing microarray analysis of miRNAs in exosomes, we tried to elucidate the details of the molecular basis for the activation of the gut–brain interaction by exosomes. Therefore, microarray analysis of miRNAs in the two types of exosomes used here, one from Uro-A-treated Caco-2 cells and the other from Uro-A-fed Aged mice, revealed the changes in miRNA expression in exosomes induced by Uro-A. Here, the whole list of miRNAs whose expression was changed upon Uro-A treatment is shown in the Appendix A. Comparing miRNAs whose expression is enhanced in exosomes derived from Uro-A-treated Caco-2 cells and miRNAs whose expression is enhanced in exosomes derived from mouse serum ingested with Uro-A, the two commonly enhanced miRNAs were miR-5100 and miR-2861. These two miRNAs may play an essential role in the functionality of Uro-A. However, there are many other miRNAs that are upregulated in each of these miRNAs, and it is necessary to analyze the functionality of each miRNA in more detail in future studies. Among the miRNAs showing expression changes in each exosome, we selected six miRNAs with the highest expression changes and estimated their target genes of the miRNAs and the pathways regulated by these target genes. The results revealed that miRNAs in Caco-2-derived and Aged-Uro-A mice-derived exosomes significantly regulated the pathways involved in memory, neurological function, and longevity (Table 1 and Table 2). In addition, compared to Caco-2-derived exosomal miRNAs, Aged-Uro-A mice-derived exosomal miRNAs were found to regulate their respective pathways more strongly and more extensively. These results suggest that exosomes, especially the miRNAs contained therein, function as part of the molecular basis for the activation of gut–brain interactions by Uro-A.

### 3.6. Involvement of Secreted Factors Other than Exosomes in the Functionality of Uro-A

In addition, we focused on other gut-derived secreted factors as the molecular basis for the activation of the gut–brain interaction by Uro-A. Therefore, the changes in gene expression in Caco-2 cells upon Uro-A treatment were examined using RNA sequencing (RNAseq) analysis, with a particular focus on secreted factors, and we traced their expression changes. The results revealed the enhanced expression of several intestine-derived secretory factors, including BDNF, NT-4, (Appendix A), and ciliary neurotrophic factor (CNTF) (Appendix A). Corresponding to the RNAseq results, this study found that Uro-A treatment significantly increased the expression of BDNF, NT-4, CNTF, and nerve growth factor (NGF) in Caco-2 cells (Figure 7A–D). For BDNF, enhanced secretion from Uro-A-treated Caco-2 cells was observed as a free protein not contained in exosomes (Figure 7E). These secretory factors, secreted in response to Uro-A, together with exosomes, are believed to form the molecular basis for the activation of the gut–brain interaction by Uro-A.

## 4. Discussion

Previous studies using mouse models of Alzheimer’s disease (AD) have shown that the activation of mitophagy by Uro-A inhibits AD related to hyperphosphorylation and prevents memory impairment in model mice and that Uro-A prevents neuronal apoptosis and enhances neurogenesis, thereby inhibiting cognitive impairment in model mice [2,3]. However, the molecular mechanism by which orally ingested Uro-A inhibits cognitive decline via the intestine remains unclear. We therefore used Aged mice as a model of age-related cognitive impairment. In this study, we demonstrated that Uro-A improved memory function in Aged mice by suppressing inflammation and inducing characteristic gene expression in the hippocampus, including gene expression similar to that in Young mice. Uro-A was shown to suppress not only memory impairment in AD model mice in previous studies but also age-related memory impairment.

Possible molecular mechanisms for food-induced activation of the gut–brain interaction include those dependent on secretory factors, including exosomes, via the vagus nerve, and gut bacteria and their products. We have been studying the possibility of the activation of the gut–brain interaction by exosomes. The results of previous studies showed that intestinal cells treated with foods such as carnosine and GABA secrete exosomes that activate neuronal cells and that exosomes derived from the serum of mice ingested with these foods also activate neuronal cells [5,13,15,32]. In this study, Uro-A-treated intestinal cell-derived exosomes, as well as Uro-A-ingested mouse serum-derived exosomes, were found to activate mitochondria in neuronal cells, indicating that Uro-A may also activate gut–brain interactions via exosomes. The results of this study also showed that mouse serum-derived exosomes are more potent in terms of neuronal cell activation. One possible reason for this is that mouse serum-derived exosomes are thought to contain exosomes from all tissues activated as a result of Uro-A ingestion and not just the intestine. This could be true because Uro-A is widely recognized for its functionality in muscles [33], and previous studies with carnosine have shown that muscle activation in particular can lead to the secretion of exosomes that can activate neuronal cells [5]. Exosomes carry a variety of factors such as proteins, DNA, and RNA and are reported to play important roles in the nervous system and neurodegenerative diseases. In particular, miRNAs in exosomes that are associated with memory formation or loss have recently been identified, and together with the results of this study, research is expected to be conducted in the future on miRNAs involved in the activation of the gut–brain interaction and the possibility of controlling brain functions through their regulation [34,35].

When discussing the possibility of neuronal mitochondria activation through gut–brain interaction activation by Uro-A, indeed, Uro-A-treated Caco-2-derived culture supernatant activated mitochondria in SH-SY5Y cells (Figure 5). In addition, both Uro-A-treated Caco-2 cell-derived exosomes and Uro-A-ingested mouse serum-derived exosomes activated mitochondria in SH-SY5Y cells (Figure 6). Furthermore, the enhanced expression of genes (SIRT1, TFAM, and Atp5d) involved in mitochondrial biosynthesis and activity in the hippocampus of Uro-A-fed mice suggests that the activation of mitochondria in neurons through the activation of gut–brain interactions by Uro-A may contribute to improvements in memory function. This is consistent with previous reports that mitochondria play an important role in the maintenance of cognitive memory [36].

Previous studies have shown that carnosine, a food that activates gut–brain interactions, acts on intestinal cells and induces the secretion of various factors, including neurotrophic factors [37]. The results showed that Uro-A induced the secretion of various neurotrophins, including BDNF, NT-4, CNTF, and NGF, from intestinal cells and that Uro-A ingestion may improve brain function by enhancing the expression of neurotrophic factors and regulating intracellular signaling molecules. BDNF is known to contribute to the gut–brain axis [38]. NT-4 regulates the survival, differentiation, and regeneration of neurons [39]. CNTF and its derived peptide reduce the age-dependent decline in learning and memory in Aged rats [40]. NGFs are a promising therapeutic target for AD [41]. Although it is not clear which is the main factor in the activation of the gut–brain interaction by Uro-A, gut-derived exosomes, and various secretory factors, it can be assumed that the activation of intestinal cells causes the secretion of various secretory factors, which ultimately activate the gut–brain interaction. Based on the results of this study, the effects of Uro-A on humans are also expected. It is also necessary to be aware of the limitations of this study, including potential biases that may exist, the limitations of using animal models, and the unavoidable imprecision associated with the results.

## 5. Conclusions

Although it is not clear which is the main factor in the activation of the gut–brain interaction by Uro-A, gut-derived exosomes, and various secretory factors, it can be assumed that the activation of intestinal cells causes the secretion of various secretory factors, which ultimately activate the gut–brain interaction.

## Figures and Tables

**Figure 1 nutrients-16-03369-f001:**
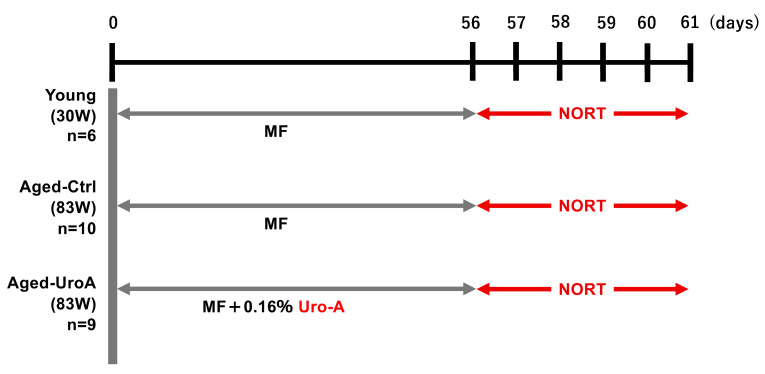
Schematic diagram of the experimental protocol.

**Figure 2 nutrients-16-03369-f002:**
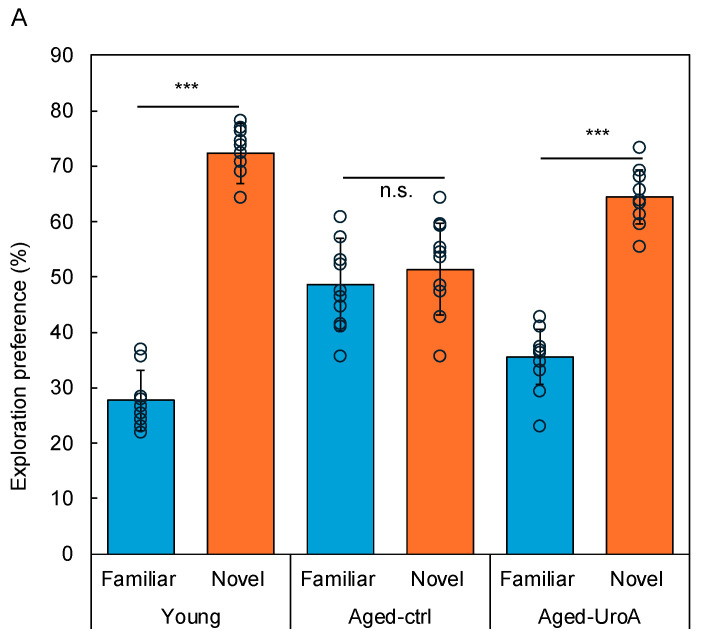
Effects of Uro-A on age-related memory impairment in Aged mice. (**A**) Comparison of exploration preference among Young, Aged-Ctrl, and Aged-Uro-A groups compared with that for the familiar object; (**B**) discrimination index compared with the Aged-Ctrl group (*** *p* < 0.001) (value means ± SEM, *n* = 10). The circles indicate the respective data. n.s. shows not significant.

**Figure 3 nutrients-16-03369-f003:**
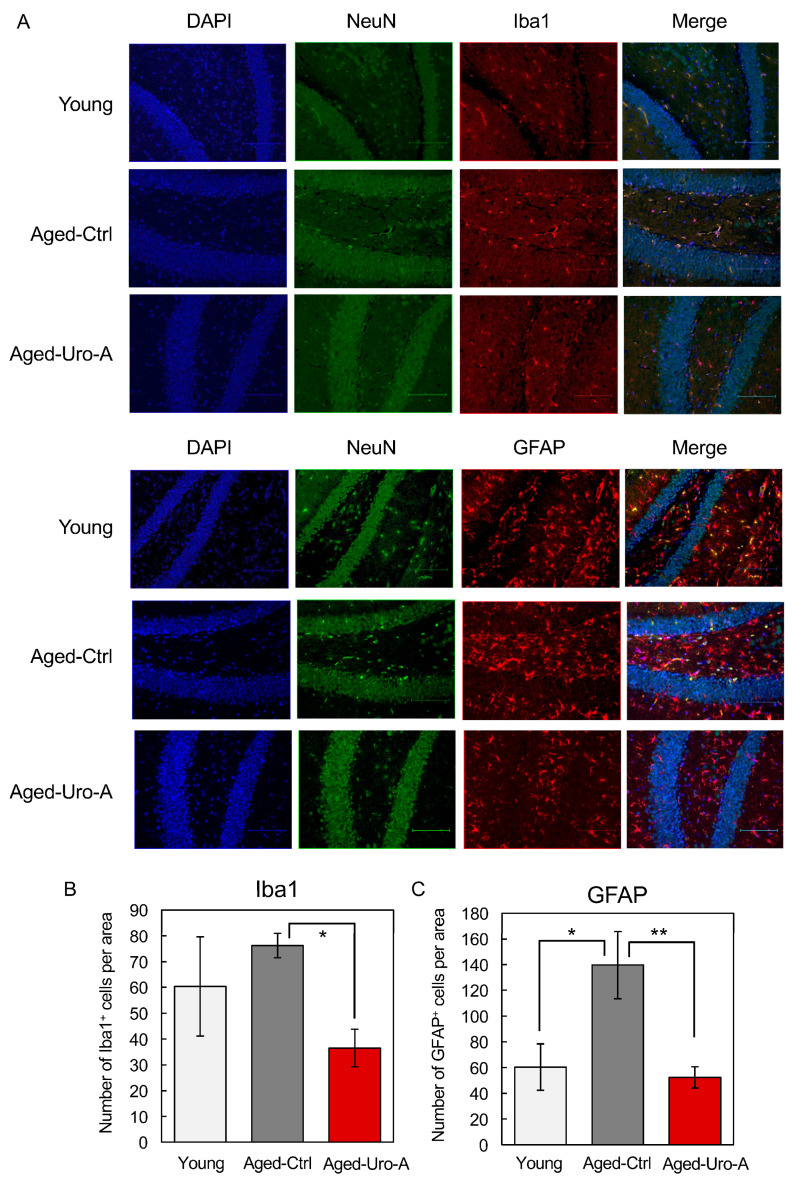
(**A**) Effects of Uro-A on age-related inflammation in Aged mice. Brain sections were incubated with primary antibodies for anti-Iba1, GFAP, and NeuN. Iba1 and GFAP were stained with Alexa Fluor 555 and NeuN with Alexa Fluor 488. After staining with Vectashield mounting medium, the tissue samples were observed under a fluorescence microscope. (**B**,**C**) Number of Iba1^+^ and GFAP^+^ cells per area, respectively (* *p* < 0.05; ** *p* < 0.01) (value means ± SEM, *n* = 10).

**Figure 4 nutrients-16-03369-f004:**
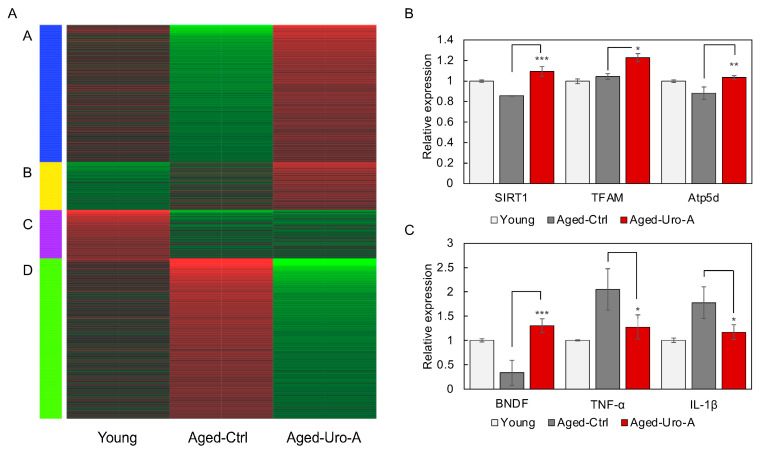
Effects of Uro-A on gene expression in the hippocampus of Aged mice. (**A**) The changes in gene expression in the hippocampus of each mouse are shown as a heatmap. The genes were classified into four clusters (A–D) according to changes in gene expression. Genes expressed at low levels are shown in green and genes expressed at high levels are shown in red. (**B**) The expression of genes (SIRT1, mitochondrial transcription factor A (TFAM), Atp5d) in the hippocampus of Uro-A-fed mice analyzed by RNAseq. (**C**) The expression of genes (BDNF, TNF-α, and IL-1β) in the hippocampus of Uro-A-fed mice analyzed by RNAseq (* *p* < 0.05; ** *p* < 0.01; *** *p* < 0.001) (value means ± SEM, *n* = 3).

**Figure 5 nutrients-16-03369-f005:**
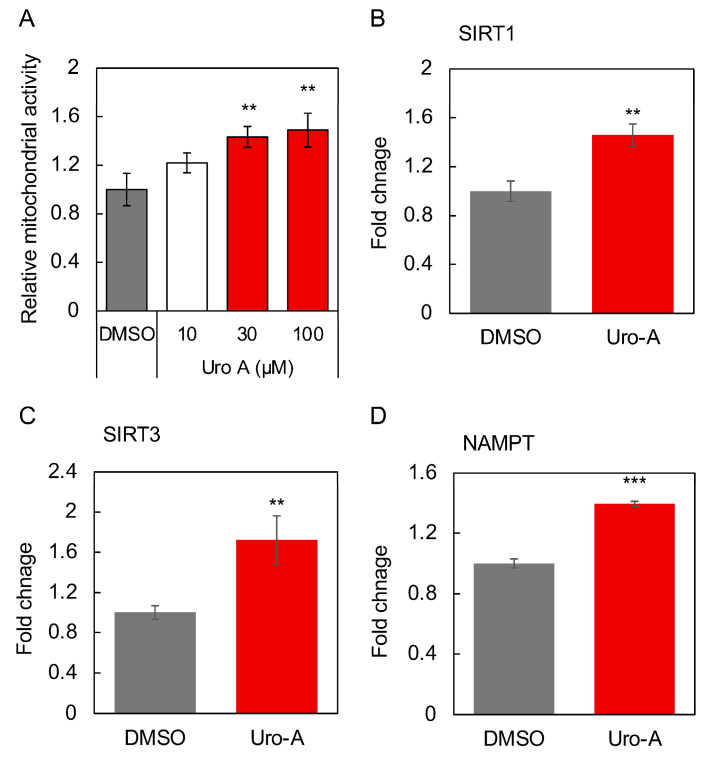
Cell supernatants of Caco-2 cells treated with Uro-A-activated SH-SY5Y cells for 2 days. Uro-A stock solution dissolved in DMSO was added at a dilution of 1/1000 to give a final concentration of 10–100 µM. In the DMSO group, the same volume of DMSO was added as at this time. DMSO-treated cells are the controls for all experiments. (**A**) Mitochondrial activity of SH-SY5Y cells treated with supernatant of Uro-A (10–100 µM)-treated Caco-2 cells. (**B**) Expression of SIRT1; (**C**) SIRT3; (**D**) NAMPT; (**E**) BDNF; and (**F**) PGC-1α in SH-SY5Y cells treated with the supernatant of 100 µM Uro-A-treated Caco-2 cells. (* *p* < 0.05; ** *p* < 0.01; *** *p* < 0.001) (value means ± SEM, *n* = 3). Experiments were repeated three times, and representative data are shown.

**Figure 6 nutrients-16-03369-f006:**
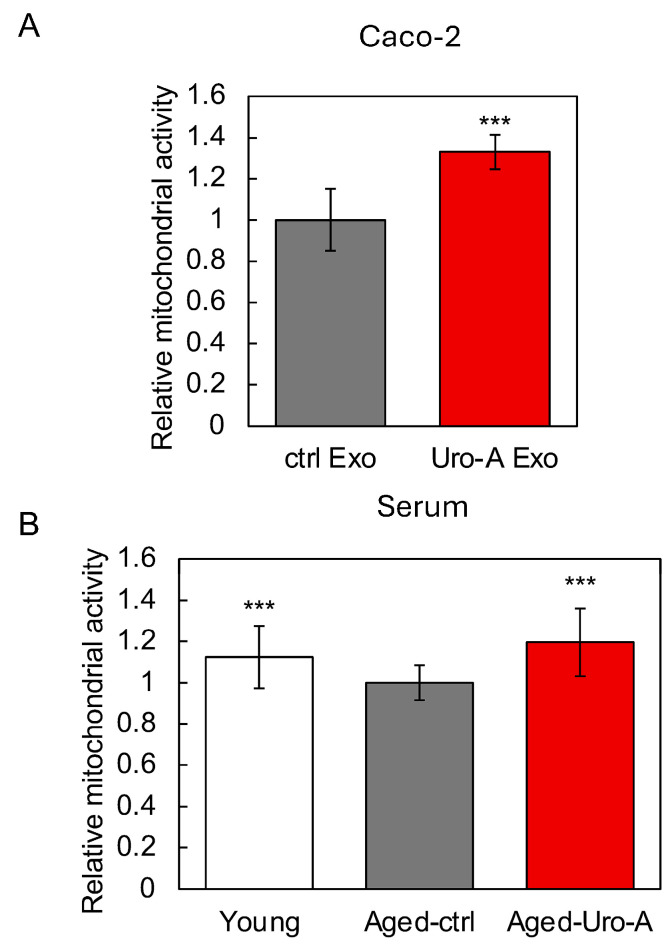
Functional evaluation of exosomes induced by Uro-A. (**A**) Effects of exosomes derived from Uro-A-treated Caco-2 cells on the activation of mitochondria; (**B**) effects of exosomes derived from serum of mice on the activation of mitochondria. (*** *p* < 0.001) (value means ± SEM, *n* = 10). Experiments were repeated three times, and representative data are shown.

**Figure 7 nutrients-16-03369-f007:**
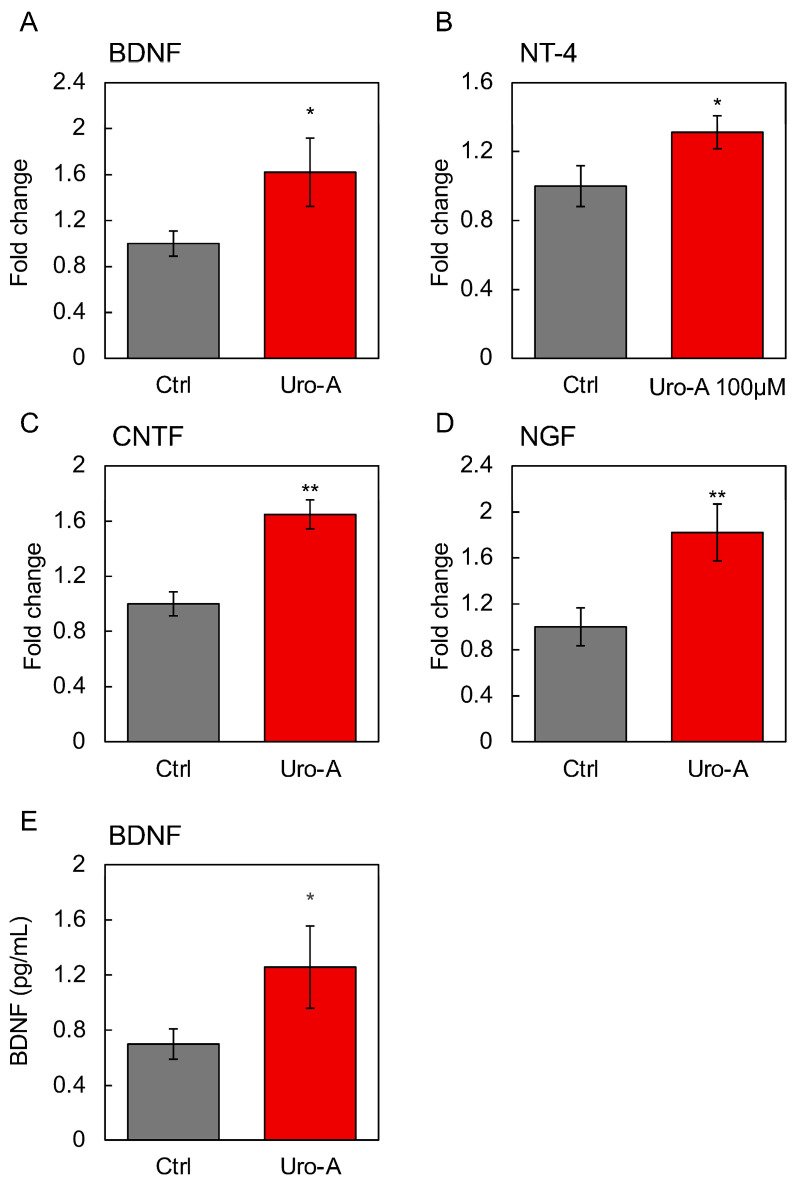
Effects of Uro-A on the gene expression of secretory factors in Caco-2 cells. Expression of (**A**) BDNF; (**B**) NT-4; (**C**) CNTF; and (**D**) NGF in Caco-2 cells treated with 100 µM Uro-A. (**E**) BDNF level in the supernatant of Caco-2 treated with Uro-A was determined by ELISA (* *p* < 0.05; ** *p* < 0.01) (value means ± SEM, *n* = 3). Experiments were repeated three times, and representative data are shown.

**Table 1 nutrients-16-03369-t001:** Pathways regulated by miRNAs derived from Uro-A-treated Caco-2 cells.

Category	Origin	Caco-2					
	miR	4730	6126	663a	4497	4745-5p	3663-3p
Memory	Axon guidance	*	***	**	*	**	***
	Long-term potentiation			*			
	Neurotrophin signaling pathway		*	***			
Neurological Function	GABAergic synapse			*			*
	Long-term depression						
	Dopaminergic synapse						**
	Oxytocin signaling pathway			***			*
Longevity Signal	Longevity regulating pathway		***	*	*		
	FoxO signaling pathway		***				
	Calcium signaling pathway	**	*		*	**	

Statistical significance was defined as *p* < 0.05 when compared to the control (* *p* < 0.05; ** *p* < 0.01; *** *p* < 0.001).

**Table 2 nutrients-16-03369-t002:** Pathways regulated by miRNAs derived from Uro-A-fed mouse serum.

Category	Origin	Mouse Serum					
	miR	29a-5p	449c-3p	6240	5100	3547-5p	2861
Memory	Axon guidance	***	***	***		***	***
	Long-term potentiation	**	**	***		***	*
	Neurotrophin signaling pathway	**	**			***	***
Neurological Function	GABAergic synapse	**	***			***	*
	Long-term depression	*	**	**		**	**
	Dopaminergic synapse	***	***			***	***
	Oxytocin signaling pathway	**	*	**		***	***
Longevity Signal	Longevity regulating pathway	**	***			***	*
	FoxO signaling pathway	***	**			**	***
	Calcium signaling pathway	***	**		**	***	***

Statistical significance was defined as *p* < 0.05 when compared to the control (* *p* < 0.05; ** *p* < 0.01; *** *p* < 0.001).

## Data Availability

The data that support the findings of this study are available from the corresponding author, Y.K., upon reasonable request. Study results can also be found in QIR (https://catalog.lib.kyushu-u.ac.jp/opac_browse/papers/?lang=0, accessed on 1 February 2024).

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
