# Peer review of "Activation of the Gut–Brain Interaction by Urolithin A and Its Molecular Basis"

_nutrients, 2024, doi:10.3390/nu16193369_

Round 1
Reviewer 1 Report
Comments and Suggestions for Authors
This study demonstrated the effects of Urolithin A (Uro-A) in improving memory recognition and neuronal mitochondrial activity, decreasing astrocyte and microglial cell activation, and changing hippocampal gene expression using mice and cell culture models. The authors utilized Caco2 cell supernatant and exosomes in the presence or absence of Uro-A to treat SH-SY5Y cells to show the gut-brain interactions, as well as using male C57BL/6J Jcl mice 30-week-old and 83-week-old to study the role of Uro-A in brain/hippocampal alterations. Biochemical and histological analyses as well as behavioral test, RNA sequencing, and miRNA microarrays were performed. Overall, the authors have shown expert skills in conducting the experiments. However, I have some major comments on the current version. The following points need to be double-checked to correct the mistakes and improve the current manuscript.
· Abstract:
o The first sentence, which serves as an introduction of Uro-A and skin epidermal cells, has no connection with the second sentence and the aim of this study. Please consider changing the introduction.
o Uro-A should be Urolithin A (Uro-A)
· Methods:
o 2.4: please indicate the number of animals used per group.
o 2.8: it is unclear which cells were used here.
o 2.10: which experiments were repeated at least 3 times?
· Results:
o The author should consider re-arranging all experimental results and changing their introductory sentences by showing the alterations in vivo and supportive information via in vitro models.
o All bar charts do not show the number of samples, so it is recommended to reflect them to make the results more intuitive and credible.
o 3.1: please describe the experimental details here or the Methods regarding:
§ Uro-A preparation and vehicle used to treat Caco2 cells;
§ the rationale for selecting 10, 30, and 100 uM;
§ how long were Caco2 cells treated with Uro-A;
§ how was Caco2 cell supernatant transferred to SH-SY5Y cells;
§ how long were SH-SY5Y cells treated with Caco2 cell supernatant
§ were SH-SY5Y cells proliferated before or during the experiment
o 3.1: it is unclear why mitochondrial activation was selected, but not neurite growth, as an indicator for brain cell activation.
o 3.1: it is unclear why SIRT3, NAMPT, BDNF, and PGC-1α were selected to study here. Also, no references were included for these genes.
o 3.1: the results of SIRT1 are in Fig. 1, but there is no description in the text.
o Fig. 1: also, there is no experimental detail on DMSO concentration and the length of treatment.
o Fig. 1: Cell viability should be considered to perform and compare the effects of Uro-A in Caco2 and SH-SY5Y cells.
o 3.2: the results of NORT alone may not conclude that “…Uro-A ingestion contributes to memory formation in aged mice by activating the gut-brain interaction”. Additional experiments related to intestinal alterations in mice should be performed to support this statement.
Fig. 2: additional information on the experimental protocol should be described.
o Line 219: please insert references for the “Previous studies” mentioned here.
o Fig. 4A: please include images of different fluorescent channels, such as, Iba1 (….color, …. nm), GFAP (….color, …. nm), DAPI (….color, …. nm), etc., followed by these merged images.
o Fig. 4A: differences in image brightness and contrast make it difficult to interpret the data.
o Fig. 4: please perform additional experiments to support neuroinflammation, e.g., measuring brain inflammatory markers, etc.
o Line 242-244: it is unclear which genes or significant genes were changed or induced specifically by Uro-A in the hippocampus. Would these only be in the neurotrophin signaling pathways?
o Line 247-248: it is unclear how BDNF and NT-3 are involved in the gut-brain interactions. Also, confirmation of these gene expressions should be performed to support the RNA-seq analysis.
o 3.5: please describe the experimental details here or the Methods regarding:
§ verification of these exosomes (that they are not other particles);
§ how exosomes were measured and used to treat the cells at which concentrations/ amounts/ volume;
§ how long were the cells treated with exosomes.
· Discussion:
o It is incorrect or a misconception to mention Alzheimer's disease (AD) and refer the animal model in this study as an AD model, as mentioned in the first paragraph.
o Line 311: it is misleading to say “Our study also aimed to determine ….” while mentioning the results of previous publications but not the current study.
o Line 322 addressed “One possible reason….”, but Line 324 indicated that “This is especially true …”. By using “This could be true…” should be more appropriate.
o Line 331: what are the functions of these genes, and please include their references.
Author Response
We are grateful to reviewer #1 for the critical comment and useful suggestion that have helped us to improve our paper. As indicated in the response that follows, we have taken the comment and suggestion into account in the revised version of our paper.
Comment #1: The first sentence, which serves as an introduction of Uro-A and skin epidermal cells, has no connection with the second sentence and the aim of this study. Please consider changing the introduction.
Response #1: In response to the comment, I revised the first sentence of Abstract.
Comment #2: Uro-A should be Urolithin A (Uro-A)
Response #2: I have revised it.
Comment #3: Methods: 2.4: please indicate the number of animals used per group.
Response #3: In response to the comment, I revised the Methods 2.4.
Comment #4: 2.8: it is unclear which cells were used here.
Response #4: In response to the comment, I revised the Methods 2.8.
Comment #5: 2.10: which experiments were repeated at least 3 times?
Response #5: For experiments that were repeated three times and representative data were shown, they are described in the figure legend.
Comment #6: Results: The author should consider re-arranging all experimental results and changing their introductory sentences by showing the alterations in vivo and supportive information via in vitro models.
Response #6: In response to the comment, we rearranged all experimental results.
Comment #7: All bar charts do not show the number of samples, so it is recommended to reflect them to make the results more intuitive and credible.
Response #7: We added the number of samples in the bar chart.
Comment #8: 3.1: please describe the experimental details here or the Methods regarding:
- Uro-A preparation and vehicle used to treat Caco2 cells;
- the rationale for selecting 10, 30, and 100 uM;
- how long were Caco2 cells treated with Uro-A;
- how was Caco2 cell supernatant transferred to SH-SY5Y cells;
- how long were SH-SY5Y cells treated with Caco2 cell supernatant
- were SH-SY5Y cells proliferated before or during the experiment
Response #8:
- The origin of Uro-A was described in 2.1.
- The rationale for selecting 10, 30, and 100 uM was described in 3.4.
- The time taken to treat Caco-2 with Uro-A has been stated in 3.4.
- In response to the comment, we revised 3.4.
- In response to the comment, we revised 3.4.
- In response to the comment, we revised 3.4.
Comment #9: 3.1: it is unclear why mitochondrial activation was selected, but not neurite growth, as an indicator for brain cell activation.
Response #9: Neurite growth was also examined at the same time, but no changes were observed. This point has also been added at 3.4.
Comment #10: 3.1: it is unclear why SIRT3, NAMPT, BDNF, and PGC-1α were selected to study here. Also, no references were included for these genes.
Response #10: In response to the comment, we revised the manuscript and added references.
Comment #11: 3.1: the results of SIRT1 are in Fig. 1, but there is no description in the text.
Response #11: In response to the comment, we revised 3.4.
Comment #12: Fig. 1: also, there is no experimental detail on DMSO concentration and the length of treatment.
Response #12: In response to the comment, we revised the legend of Fig. 5 (old Fig. 1).
Comment #13: Fig. 1: Cell viability should be considered to perform and compare the effects of Uro-A in Caco2 and SH-SY5Y cells.
Response #13: In response to the comment, we revised 3.4.
Comment #14: 3.2: the results of NORT alone may not conclude that “…Uro-A ingestion contributes to memory formation in aged mice by activating the gut-brain interaction”. Additional experiments related to intestinal alterations in mice should be performed to support this statement.
Response #14: In response to the comment, we revised 3.1.
Comment #15: Fig. 2: additional information on the experimental protocol should be described.
Response #15: It was noted that the detailed protocol is explained in the Materials and Methods. Figure 1 (old Fig. 2) were also amended accordingly.
Comment #16: Line 219: please insert references for the “Previous studies” mentioned here.
Response #16: In response to the comment, we added references in 3.2.
Comment #17: Fig. 4A: please include images of different fluorescent channels, such as, Iba1 (….color, …. nm), GFAP (….color, …. nm), DAPI (….color, …. nm), etc., followed by these merged images.
Response #17: In response to the comment, we reworked our image data in new Fig. 3A.
Comment #18: Fig. 4A: differences in image brightness and contrast make it difficult to interpret the data.
Response #18: In response to the comment, we have prepared images with matching brightness and contrast.
Comment #19: Fig. 4: please perform additional experiments to support neuroinflammation, e.g., measuring brain inflammatory markers, etc.
Comment #19: In response to the comment, we added the result on expression of neuroinflammation marker (TNF-α and IL-1β) in Fig. 4C based on RNAseq results.
Comment #20: Line 242-244: it is unclear which genes or significant genes were changed or induced specifically by Uro-A in the hippocampus. Would these only be in the neurotrophin signaling pathways?
Response #20: In response to the comment, we added the results on expression of genes involved in mitochondria biogenesis and activity (SIRT1, TFAM, Atp5d), neurotrophin (BDNF) and neuroinflammation marker (TNF-α and IL-1β) in Fig. 4B and C based on RNAseq results..
Comment #21: Line 247-248: it is unclear how BDNF and NT-3 are involved in the gut-brain interactions. Also, confirmation of these gene expressions should be performed to support the RNA-seq analysis.
Response #21: As the reviewer stated, the involvement of BDNF and NT-3 in the gut-brain interaction cannot be inferred from this data, so we have removed this part. The results of RNAseq analysis of BDNF and NF-3 expression changes have been added as Supplemental Figure (Figure S1).
Comment #22: 3.5: please describe the experimental details here or the Methods regarding:
- verification of these exosomes (that they are not other particles);
- how exosomes were measured and used to treat the cells at which concentrations/ amounts/ volume;
- how long were the cells treated with exosomes.
Response #22:
1)-3): In response to the comment, we revised 2.8.
Comment #23: Discussion: It is incorrect or a misconception to mention Alzheimer's disease (AD) and refer the animal model in this study as an AD model, as mentioned in the first paragraph.
Response #23: In response to the comment, we revised the first paragraph of Discussion.
Comment #24: Line 311: it is misleading to say “Our study also aimed to determine ….” while mentioning the results of previous publications but not the current study.
Response #24: In response to the comment, we revised second paragraph of Discussion.
Comment #25: Line 322 addressed “One possible reason….”, but Line 324 indicated that “This is especially true …”. By using “This could be true…” should be more appropriate.
Response #25: In response to the comment, we revised the manuscript.
Comment #26: Line 331: what are the functions of these genes, and please include their references.
Response #26: In response to the comment, we revised the Discussion.
Reviewer 2 Report
Comments and Suggestions for Authors
The manuscript by Kubota D et al., titled “Activation of the gut-brain interaction by Uro-A and its molecular basis,” deals with the anti-inflammatory effect of Uro-A on gut-brain interaction linked to memory. Uro-A mediated the activation of gut-brain interaction by exosomal miRNA and secretory key molecules (BDNF, NT-4, CNTF and NGF). Oral sub-ministration of Uro-A in the mice presents positive effects, improving brain functions and preventing memory impairment in aged mice. Therefore, the authors suggest that Uro-A is a promising therapeutic agent in the treatment of hippocampal memory system issues. The presented topic is interesting, but the study requires further analysis before being accepted.
The main issue concerning the manuscript (see comments of the discussion section) is the shallow elaboration of the different set of the in-vitro experiments with in vivo-ones while it should be important to establish their connection. A better characterization of the molecular pathways involved in the effect of Uro-A is key to substantiating the effects observed in vivo as well as it would be helpful to develop a gut-brain axis model on a chip using human gut epithelial cells (Caco-2; HT29) and a neuroblastoma cell line (SH-SY5Y, SK-N—SK-N-SH and SK-N-BE) to further validate the results obtained in the early experimental section and relate them to the in vivo study to significantly improve the understanding of these pathways in the gut axis.
SPECIFIC COMMENTS
Material and Methods section
- Paragraph 2.1: No indication in the experimental procedure followed during the Uro-A-stimulation of Caco2 and Caco2-supernatant stimulation of SH-SY5Y are given on the time incubation, cell number, and the chosen criteria for the Caco2-supernatant amount in relation to the SH-SY5Y number.
- Paragraph 2.4: The authors should explain why they used only male mice and not female ones. Besides, they should approximately indicate the amount (mg) of Uro-A per Kg mouse body weight eaten by the mice
Results Section
- Paragraph 3.1: Excluding data on mitochondrial activity, all the results regarding gene expression lack data on cells non-supplemented with DMSO as a control in addition to DMSO-treated cells in their assays. Is the fold-increase calculated on control without DMSO. This nedd to be better clarified.
Besides, although Uro-A is rapidly metabolized by cell lines, the authors should truly demonstrate that the Caco-2 supernatant effects on neuroblastoma cell lines are completely independent of Uro-A and then that Uro-A is completely absent in the supernatant. A line guide could be the evaluation of AhR signalling pathways in the presence or absence of Uro-A antagonists or another noteworthy methodology.
- Paragraph 3.3: The interpretation of results on Iba1 in response to Uro-A in Aged-UroA mice is conflicing. Recent studies have demonstrated that Iba1-positive microglia contribute to objective recognition memory (DOI: 10.1073/pnas.2115539118). Then the reduction of Iba1-positive microglia might indicate a reduced activation but also results in microglia memory functional impairment. Moreover, microglia shift through different profiles towards reactive or amoeboid states associated with different functions, particularly during aging, they undergo specific morphological transitions, exhibiting an inflammatory hypersensitive phenotype. Therefore, markers present specifically on the microglia of the aged brain should be used. Besides, the microglia’s functional state, depending on microglia-related gene expressions, could be detected by gene expression analysis (qPCR) helping in shifts in the microglia population towards active or amoeboid states. These aspects need to be discussed more deeply in the manuscript. Finally, to support the role of Uro-A as a potent mediator in reducing hippocampal inflammation, the detection of the expression of neuroinflammatory markers, including pro-inflammatory cytokines, should be preferred.
- Paragraph 3.4: The heatmap should indicate the single sample in each subgroup, and the A-gene and B-gene groups should be presented as a list in the supplementary.
- Paragraph 3.5: The microarray analysis of miRNA should be improved by presenting a list of whole miRNAs found in the presented conditions (control and Uro-A stimulation) in a panel indicating miRNA relative expression (median range and DCt) and p-value comparison.
Moreover, the comparison of the relative miRNA expression levels between Uro-A-stimulation Caca2 and Uro-A-fed aged mice and their related controls (not-treated Caca2 and aged mice) need to be better explained in the test
- Paragraph 3.6: Since gene expression is controlled by several levels, it should be better to analyze the effects of Uro-A on the gene expression of secretory factors (BDNF, NT-4, CNTF, and NGF) at the translation level by measuring the proteins in the Caco2 supernatant and in the mouse serum in response to Uro-A. In particular, it is necessary to evaluate if the mentioned proteins are free-secreted or included in the exosome in the supernatant.
Discussion Section
The section does not analyze all the results of the study. No efforts have been maid to connect the results of in-vitro experiments with in vivo-ones. The connection with Carnosine and GABA diets that have not been analyzed by the authors is confusing. Moreover, some concepts are repeated more times without going deeper into the relationship between the discovered miRNAs and their target pathways involved in memory performance. In addition, it is missing the inclusion of summarized previous studies on miRNA-dependent signalling pathways modulating spatial learning and memory capability.
What’s more, it must more thoroughly imply the implications of mitochondria for cognitive function and memory decline, indicating which of the found genes are linked to the mitochondrial signalling pathway and supporting all of that with previous studies.
Further, the limitation of the study are lacking

Comments on the Quality of English LanguageThe quality of English is adequate
Author Response
Reviewer #2
We are grateful to reviewer #2 for the critical comment and useful suggestion that have helped us to improve our paper. As indicated in the response that follows, we have taken the comment and suggestion into account in the revised version of our paper.
Comment #1: The main issue concerning the manuscript (see comments of the discussion section) is the shallow elaboration of the different set of the in-vitro experiments with in vivo-ones while it should be important to establish their connection. A better characterization of the molecular pathways involved in the effect of Uro-A is key to substantiating the effects observed in vivo as well as it would be helpful to develop a gut-brain axis model on a chip using human gut epithelial cells (Caco-2; HT29) and a neuroblastoma cell line (SH-SY5Y, SK-N—SK-N-SH and SK-N-BE) to further validate the results obtained in the early experimental section and relate them to the in vivo study to significantly improve the understanding of these pathways in the gut axis.
Response #1: As the reviewer states, the proposed system would also enable us to obtain more detailed information on the molecular basis of the in vivo functionality of Uro-A. We realised that the contextual flow presented in this manuscript made it difficult to understand and confusing to explain the molecular basis that could explain the in vivo results, we have significantly modified the structure of our results, first presenting studies on the in vivo function of Uro-A and then analysing the molecular basis of its functionality.
Comment #2: Paragraph 2.1: No indication in the experimental procedure followed during the Uro-A-stimulation of Caco2 and Caco2-supernatant stimulation of SH-SY5Y are given on the time incubation, cell number, and the chosen criteria for the Caco2-supernatant amount in relation to the SH-SY5Y number.
Response #2: In response to the comment, we revised 3.4.
Comment #3: Paragraph 2.4: The authors should explain why they used only male mice and not female ones. Besides, they should approximately indicate the amount (mg) of Uro-A per Kg mouse body weight eaten by the mice
Response #3: For behavioural tests with aged mice, two papers were referred to. As male mice were used in both of these papers, male mice were also used in our study. This point is also discussed in the paper.
- Brain, 9: 11, 2016 : Age-related changes in behavior in C57BL/6J mice from young adulthood to middle age
- , 39: 100-118 (2019): Age-related behavioral changes from young to old age in male mice of a C57BL/6J strain maintained under a genetic stability program
Comment #4: Paragraph 3.1: Excluding data on mitochondrial activity, all the results regarding gene expression lack data on cells non-supplemented with DMSO as a control in addition to DMSO-treated cells in their assays. Is the fold-increase calculated on control without DMSO. This need to be better clarified.
Response #4: In Fig. 5A (old Fig. 1A), we would confuse the reviewer. In Fig. 5 (old Fig. 1), DMSO-treated cells are the controls for all experiments. Accordingly, the Fig. 5A (old Fig. 1A) has been corrected and the legend has also been amended.
Comment #5: Besides, although Uro-A is rapidly metabolized by cell lines, the authors should truly demonstrate that the Caco-2 supernatant effects on neuroblastoma cell lines are completely independent of Uro-A and then that Uro-A is completely absent in the supernatant. A line guide could be the evaluation of AhR signalling pathways in the presence or absence of Uro-A antagonists or another noteworthy methodology.
Response #5: The residual Uro-A in the Caco-2 culture supernatant after treatment with Uro-A for 2 days was confirmed by LC-MS to be below the detection limit. This point has been added in 3.4.
Comment #6: Paragraph 3.3: The interpretation of results on Iba1 in response to Uro-A in Aged-UroA mice is conflicing. Recent studies have demonstrated that Iba1-positive microglia contribute to objective recognition memory (DOI: 10.1073/pnas.2115539118). Then the reduction of Iba1-positive microglia might indicate a reduced activation but also results in microglia memory functional impairment. Moreover, microglia shift through different profiles towards reactive or amoeboid states associated with different functions, particularly during aging, they undergo specific morphological transitions, exhibiting an inflammatory hypersensitive phenotype. Therefore, markers present specifically on the microglia of the aged brain should be used. Besides, the microglia’s functional state, depending on microglia-related gene expressions, could be detected by gene expression analysis (qPCR) helping in shifts in the microglia population towards active or amoeboid states. These aspects need to be discussed more deeply in the manuscript. Finally, to support the role of Uro-A as a potent mediator in reducing hippocampal inflammation, the detection of the expression of neuroinflammatory markers, including pro-inflammatory cytokines, should be preferred.
Response #6: Thank you for your very helpful comments. In response to the comment, the involvement of Iba1-positive microglia is discussed and added to 3.2, together with results on the change in expression of neuroinflammation markers.
Comment #7: Paragraph 3.4: The heatmap should indicate the single sample in each subgroup, and the A-gene and B-gene groups should be presented as a list in the supplementary.
Response #7: The heatmap shows the results of the analysis of RNA from three mice mixed together for each subgroup. We will add a note on this point. As for the heat maps, they were re-analysed and the pathways involving genes belonging to each cluster were organized and added as Table S1.
Comment #8: Paragraph 3.5: The microarray analysis of miRNA should be improved by presenting a list of whole miRNAs found in the presented conditions (control and Uro-A stimulation) in a panel indicating miRNA relative expression (median range and DCt) and p-value comparison. Moreover, the comparison of the relative miRNA expression levels between Uro-A-stimulation Caco-2 and Uro-A-fed aged mice and their related controls (not-treated Caco-2 and aged mice) need to be better explained in the test.
Response #8: In response to the comment, we have listed the whole miRNAs whose expression were changed upon Uro-A treatment, and showed in Supplemental Figure (Table S2 and S3), and revised 3.5.
Comment #9: Paragraph 3.6: Since gene expression is controlled by several levels, it should be better to analyze the effects of Uro-A on the gene expression of secretory factors (BDNF, NT-4, CNTF, and NGF) at the translation level by measuring the proteins in the Caco2 supernatant and in the mouse serum in response to Uro-A. In particular, it is necessary to evaluate if the mentioned proteins are free-secreted or included in the exosome in the supernatant.
Response #9: In response to the comment, we have added data on BDNF secreted from Uro-A-treated Caco-2 cells and revised 3.6.
Comment #10: The section does not analyze all the results of the study. No efforts have been paid to connect the results of in-vitroexperiments with in vivo-ones. The connection with Carnosine and GABA diets that have not been analyzed by the authors is confusing. Moreover, some concepts are repeated more times without going deeper into the relationship between the discovered miRNAs and their target pathways involved in memory performance. In addition, it is missing the inclusion of summarized previous studies on miRNA-dependent signalling pathways modulating spatial learning and memory capability.
Response #10: The order of the results has been changed to make the relationship between in vivo and in vitro experiments easier to understand. The sentences on carnosine and GABA were confusing and has been revised. We have also added a note on a paper showing the potential for learning and memory regulation by miRNAs in exosomes.
Comment #11: What’s more, it must more thoroughly imply the implications of mitochondria for cognitive function and memory decline, indicating which of the found genes are linked to the mitochondrial signalling pathway and supporting all of that with previous studies.
Response #11: In response to the comment, we added Fig. 4B, and discussed in the discussion section.
Comment #12: Further, the limitation of the study are lacking
Response #12: In the revised version, we added the limitation of the study at discussion.
Round 2
Reviewer 1 Report
Comments and Suggestions for Authors
Overall, the authors have addressed all the comments and performed additional analyses accordingly. There are some minor concerns to be considered as follows:
n Please adjust the image clarity of Fig. 3A and 3B.
n Please consider including individual data with symbols (such as circles, squares, triangles, etc.) to all bar charts.
Author Response
We are grateful to reviewer #1 for the comments and useful suggestion. In response to the comments, we revised the manuscript.
Comment #1:
Please adjust the image clarity of Fig. 3A and 3B.
Response #1:
As you can see in Fig. 3A and B, the brightness of the stained images of Iba1 and GFAP are very different, and we have adjusted the brightness of the two images as much as possible in the revised version. We hope you understand.
Comment #2:
Please consider including individual data with symbols (such as circles, squares, triangles, etc.) to all bar charts.
Response #2:
In response to the comment, data symbols were added to the bar chart in Fig. 2, which has a relatively large number of N.
Considering that data symbols are generally not added to bar charts of qPCR and ELISA for triplicate data, data symbols are not added in these bar charts. For mitochondria data, on the other hand, data symbols are not added because the number of N is too large. Thank you for your understanding.
Reviewer 2 Report
Comments and Suggestions for Authors
The authors have satisfactorily addressed all my issues, improving the manuscript. I do not have any other concerns regarding it, so I would recommend it for publication.
Author Response
We are grateful to reviewer #2 for the comments and useful suggestions.